# Influence of Microbial Metabolites and Itaconic Acid Involved in Bacterial Inflammation on the Activity of Mitochondrial Enzymes and the Protective Role of Alkalization

**DOI:** 10.3390/ijms23169069

**Published:** 2022-08-14

**Authors:** Nadezhda Fedotcheva, Natalia Beloborodova

**Affiliations:** 1Institute of Theoretical and Experimental Biophysics, Russian Academy of Sciences, Institutskaya Str., 3, 142290 Pushchino, Moscow Region, Russia; 2Federal Research and Clinical Center of Intensive Care Medicine and Rehabilitology, 25-2 Petrova Str., 107031 Moscow, Moscow Region, Russia

**Keywords:** mitochondrial dysfunction, inflammation, itaconic acid, microbial metabolites, acidosis, succinate, glutamate, biomarkers

## Abstract

Human microbiota produces metabolites that may enter the bloodstream and exert systemic influence on various functions including mitochondrial. Mitochondria are not only a target for microbial metabolites, but also themselves, due to the inhibition of several enzymes, produce metabolites involved in infectious processes and immune response. The influence of indolic acids, microbial derivatives of tryptophan, as well as itaconic acid, formed in the tricarboxylic acid cycle under the action of bacterial lipopolysaccharides, on the activity of mitochondrial enzymes was studied by methyl thiazolyl tetrazolium (MTT), dichlorophenolindophenol (DCPIP) and pyridine nucleotide fluorescence assays. Thus, it was found that indolic acids suppressed succinate and glutamate oxidation, shifting the redox potential of pyridine nucleotides to a more oxidized state. Itaconic acid, in addition to the well-known inhibition of succinate oxidation, also decreased NAD reduction in reactions with glutamate as a substrate. Unlike itaconic acid, indolic acids are not direct inhibitors of succinate dehydrogenase and glutamate dehydrogenase as their effects could be partially eliminated by the thiol antioxidant dithiothreitol (DTT) and the scavenger of lipid radicals butyl-hydroxytoluene (BHT). Alkalization turned out to be the most effective means to decrease the action of these metabolites, including itaconic acid, which is due to the protective influence on redox-dependent processes. Thus, among mitochondrial oxidative enzymes, the most accessible targets of these microbial-related metabolites are succinate dehydrogenase and glutamate dehydrogenase. These are important in the context of the shifting of metabolic pathways involved in bacterial inflammation and sepsis as well as the detection of new markers of these pathologies.

## 1. Introduction

It is well known that human microbiota produces metabolites that may enter the bloodstream and exert systemic effects on various functions in both healthy and pathological states. Numerous data show that microbial metabolites are involved in the regulation of the immune system [1], the central nervous system [2], metabolism, and epigenetic control [3,4].

Mitochondria are one of the targets of microbial metabolites. Their participation in the development of mitochondrial dysfunction was found in pathologies associated with infections and sepsis [5,6]. It is especially important that they by themselves produce metabolites involved in the infectious process and immune response. First of all, this refers to a specific metabolite, itaconic acid, which is absent in the norm and is formed in the tricarboxylic acid cycle only under the action of bacterial lipopolysaccharides. Itaconic acid is synthesized in immune blood cells by cis-aconitate decarboxylase, the biosynthesis of which is induced in response to an infection [7,8]. Itaconic acid activates the production of anti-inflammatory cytokines and, being an inhibitor of the glyoxylate cycle, has an antimicrobial effect on microorganisms in which this cycle is present. Moreover, being a competitive inhibitor of succinate dehydrogenase, itaconic acid changes mitochondrial and cellular metabolism [7].

Along with itaconic acid, conventional metabolites of the tricarboxylic acid cycle are currently considered as immunomodulators [9,10]. These include succinate, α-ketoglutarate, fumarate, and citrate, for which the role in the immunometabolism and inflammation was estimated [9,10,11]. Although these metabolites are generally associated with bioenergetic and biosynthetic processes, they also control or modulate the immune response, activating the immune cells and the production of pro- or anti-inflammation cytokines. In this context, the study of the role of microbial metabolites, along with bacterial lipopolysaccharides, in the modulation of the activity of mitochondrial enzymes is of fundamental importance.

We have previously shown the involvement of microbial phenolic acids, the metabolites of phenylalanine, in the development of the mitochondrial dysfunction. The concentrations of some of them increased during infections and sepsis, and correlated with a rise in the levels of mitochondrial dicarboxylic acids and lactate in the blood [5]. In this work, we studied the effect of indole acids, the metabolites of the microbial pathway of tryptophan degradation, on the activity of mitochondrial oxidative enzymes. It is known that indole derivatives of tryptophan are formed with the direct participation of the intestinal microbiota and include tryptamine, indole, 2-oxoindole, indole-3-carboxylate, indole-3-acetate, indole-3-propionate, indole-3-lactate, etc. [12,13]. These indole metabolites are involved in systemic homeostasis along the so-called microbiota-gut-brain and microbiota-gut-liver axes; some of them use aryl-hydrocarbon receptors as the main target of interactions with the host organism [14]. It is also known that various tryptophan derivatives are adsorbed on the intestinal epithelium and enter the bloodstream, influencing various organs, tissues, and physiological functions [15]. With regard to their action on mitochondria, it was shown that indole-3-acetate decreases the efficiency of oxidative phosphorylation, declines the rate of uncoupled respiration, and inhibits the complexes of the respiratory chain [16].

In this work, we examined the effect of indole-3-carboxylate, also known as 3-carboxyindole or 3-indolylformic acid (ICA), indole-3-acetate (IAA), and indole-3-lactate (ILA), the microbial origin of which has been proven, on the activity of mitochondrial dehydrogenase and some mitochondrial functions. We also compared their effects with the action of itaconic acid, as an inhibitor of succinate dehydrogenase, with an emphasis on the role of acidification and alkalization in these processes.

## 2. Results

### 2.1. Influence of Indolic Acids on the Oxidation of Mitochondrial Substrates Measured by the MTT Assay

Figure 1 demonstrates the influence of indolic acids on the activity of the oxidation of succinate and NAD-dependent substrates when tested on isolated rat liver mitochondria by the MTT assay. These compounds exerted a different intensity effect on succinate oxidation (Figure 1a). The specificity of the measurement of the succinate oxidation activity is evidenced by the inhibition of MTT reduction by malonate, a selective succinate dehydrogenase inhibitor. In the range of concentrations from 50 to 500 μM, the reduction of MTT was inhibited to a greater extent in the presence of ICA, to a lesser extent with IAA, and in the presence of ILA no inhibition was observed. None of them at the same concentrations affected the oxidation of glutamate plus malate (Figure 1b). Considering that ICA was the most active indole metabolite, we used it in all subsequent experiments.

When testing with each NAD-dependent substrate, it turned out that ICA causes the inhibition not only of succinate oxidation but also of glutamate oxidation (Figure 1c). At a concentration of 200 μM, ICA decreased the oxidation of succinate and glutamate by 40–50% and only slightly affected the oxidation of all other NAD-dependent substrates. The inhibition of MTT reduction during glutamate oxidation depended on ICA concentration (Figure 1d). Also, the inhibition by malonate shows that a certain contribution to MTT reduction by glutamate was made by the oxidation of succinate, probably both endogenous and generated during the oxidation of glutamate (Figure 1d).

### 2.2. Influence of Itaconic Acid on the Oxidation of Mitochondrial Substrates, as Measured by the MTT Reduction

We compared the influence of ICA on the oxidation of mitochondrial substrates with the impact of ITA. ITA is known to inhibit succinate oxidation and does not influence the oxidation of glutamate plus malate, or pyruvate plus malate [7]. As shown in Figure 2, ITA inhibited the oxidation of succinate and did not affect the oxidation of glutamate plus malate as measured by the MTT assay. However, like ICA, it also inhibited glutamate oxidation. The inhibition of MTT reduction during glutamate oxidation depended on ITA concentration (Figure 2b). Thus, the influence of ITA almost replicated the influence of ICA. An essential difference between them is that ITA acts at concentrations of 2–4 mM, while ICA produces the effect already at a concentration of 200–500 μM.

### 2.3. Influence of Itaconic and Indolcarboxylic Acids on the Oxidation of NAD-Dependent Substrates

The influence of ICA and ITA on NADH oxidation was examined by a fluorescent assay. 

As shown in Figure 3a, in the presence of ITA or ICA and glutamate, the rate of NADH oxidation increased several times. Figure 3b shows that the stimulation of NADH oxidation is not associated with an uncoupling and a drop in the membrane potential, while at the same concentrations, the substances only had a very weak effect on the membrane potential maintained by glutamate oxidation (Figure 3b). However, the activation of NADH oxidation was removed by rotenone, an inhibitor of the first complex of the respiratory chain (Figure 3c,d).

Inhibition of NADH oxidation by rotenone was complete in the case of ITA. In the presence of ICA, rotenone inhibited NADH oxidation by about 75%; consequently, the residual part of the NADH diminution is not associated with the complex I of the respiratory chain. All other NAD-dependent substrates responded differently to the addition of ITA and ICA; namely, the redox state of NADH supported by glutamate with malate was resistant to the action of both. It was decreased in the presence of KGL or malate, but to a lesser extent than in the case of the glutamate supplementation (Figure 3e,f). Thus, both metabolites activated NADH oxidation predominantly in the case of the oxidation of glutamate in the comparison with other NAD-dependent substrates, which may be associated with the inhibition of glutamate dehydrogenase (GDH) itself. Indeed, when testing these metabolites on permeabilized (for NAD penetration) mitochondria, both decreased the rate and the amplitude of NAD reduction (Figure 4). With an increase in concentration from 2 to 8 mM, ITA decreased the amplitude and the rate of NAD reduction in the range from 20 to 60% (Figure 4a,b). ICA had a much weaker effect, causing a 20 and 30% inhibition at concentrations of 250 and 500 μM, respectively (Figure 4c,d).

### 2.4. Influence of Itaconic and Indolcaboxylic Acids on the Activity of SDH, as Measured by DCPIP Reduction

Similar differences were observed in the experiments in which the influence of these metabolites on succinate dehydrogenase (SDH) activity was examined using the specific acceptor DCPIP. As expected, ITA significantly inhibited succinate oxidation, and this inhibition was not eliminated by the intermediate electron carrier PMS, indicating the inhibition of the catalytic subunit of the enzyme (Figure 5a). The inhibition reached 50% at an ITA concentration of 2 mM. ICA, in contrast, had no effect on SDH activity (Figure 5a). The acidification of the incubation medium to pH 6.7 significantly increased the inhibition of SDH by ITA. Under these conditions, the inhibition reached 70% as compared with the control values at neutral pH, while ICA still had no effect on the enzyme activity.

Whereas acidification enhances the action of ITA-on-SDH activity, alkalization can produce an opposite effect, since, according to our data, alkalization promotes the restoration of mitochondrial functions under the influence of some damaging agents [17,18]. As shown on Figure 5c, the alkalization of the medium to pH 8.0 significantly decreased the inhibition of SDH by ITA, had no influence on the effect of ICA, and sharply activated the reduction of DCPIP by PMS in all cases. Figure 5d summarizes the data on the inhibition of the activity of SDH by ICA and ITA depending on the pH values. Thus, ICA had no direct effect on SDH at any pH values; conversely, ITA caused the inhibition of the enzyme which increased with acidification. The influence of malonate and TTFA, the inhibitors of catalytic and ubiquinone-binding subunits of the enzyme, also varied with pH, especially in the case of the activation by PMS. As expected the inhibition by malonate was not removed by PMS at all pH values (Figure 5e,f). However, the elimination of the TTFA-induced inhibition was strongly dependent on pH and was the strongest at pH 8 and the lowest at pH 6.7 (Figure 5e). These data show that the possibilities for the recovery of SDH activity decreases under conditions of acidosis and increases with alkalization. This conclusion was confirmed in the following experiments aimed at evaluating the role of alkalization in the reduction of MTT and NAD in the presence of ITA or ICA.

### 2.5. Protective Role of Alkalization against the Influence of Itaconic and Indolcaboxylic Acids on the Activity of Mitochondrial Enzymes

Unlike DCPIP, the MTT reduction supported by succinate oxidation was suppressed by ICA, as was shown above. An essential difference between these acceptors is that DCPIP is used only in lysed mitochondria, while MTT is effective in intact organelles. The influence of ITA and ICA on MTT reduction under conditions of acidosis and alkalization is shown in Figure 6a,b. It can be seen that the inhibition by ICA and ITA increased upon acidification and was almost completely eliminated upon alkalization. Moreover, this tendency appeared both during succinate and glutamate oxidation. Compared to neutral pH, acidification to pH 6.7 resulted in an almost twofold increase in inhibition; alkalization to pH 8.0, on the contrary, almost completely prevented inhibition. This effect was especially pronounced in the case of ICA supplementation.

How alkalization affects the redox state of NADH during glutamate oxidation in the presence of ITA and ISA is shown on Figure 6c,d. While both metabolites induced NADH oxidation at normal pH, the NADH redox potential was maintained at a more stable level in the case of alkalization (Figure 6c). The maintenance of the redox state made it possible to evaluate their effect on the oxidative phosphorylation. In these conditions ITA had a significant effect on the redox state of NADH during oxidative phosphorylation, which manifested itself in a more than twofold deceleration and an incomplete reduction of NAD after ADP phosphorylation compared to the control (Figure 6d). ICA also inhibited, even to a greater extent than ITA, NAD reduction after ADP phosphorylation. Thus, alkalization prevented the oxidation of NADH, induced by ITA and ICA, but did not support NAD reduction in the course of oxidative phosphorylation. The addition of an uncoupler (FCCP) shows a complete oxidation of pyridine nucleotides in every case. Thus, ITA and ICA disturb the restoration of the redox state of pyridine nucleotides during oxidative phosphorylation.

We also examined the possible protective role of other factors that, according to our previous data, reduce the effect of aromatic microbial metabolites on mitochondria [19]. For this purpose, the lipid radical scavenger BHT and the thiol antioxidant DTT were used. BHT attenuated NADH oxidation induced by ICA or ITA, the effect being more efficient in the case with ICA (Figure 7a). Also, BHT partially eliminated the ICA-induced inhibition of MTT reduction during succinate oxidation and did not affect it in the presence of ITA (Figure 7b). The effects of both metabolites were partially removed by DTT, which indicates the participation of thiol groups in the inhibition of succinate oxidation.

## 3. Discussion

Thus, our study showed that indolic acids influenced the activity of mitochondrial dehydrogenases. Their targets were succinate dehydrogenase and glutamate dehydrogenases, since only the oxidation of their substrates was subjected to alterations in the presence of indolic acids when tested by various methods. According to our data, ITA had a similar effect on the oxidation of these substrates. Namely, in addition to the well-known inhibition of succinate oxidation, ITA activated NADH oxidation and decreased NAD reduction when glutamate served as a substrate. Unlike ITA, indolic acids are not direct inhibitors of SDH, which follows from our data with the acceptor DCPIP, which is specific to this enzyme. The inhibition of succinate oxidation by indolic acids was only observed in intact mitochondria, which indicates the involvement of other concomitant factors. These can include redox-dependent reactions, since the effects of indolic acids were partially removed by the thiol antioxidant DTT and the scavenger of lipid radicals BHT.

Itaconic acid, like indolic acids, can be considered as a microbiota-associated metabolite, since it is formed in mitochondria only in response to bacterial LPS, a component of the outer membrane of Gram-negative bacteria. The indole derivatives of tryptophan indole-3-carboxylate, indole-3-acetate, and indole-3-lactate are formed with the direct participation of the intestinal microbiota. It is known that tryptophan catabolism is mediated by several metabolic pathways, leading to the formation of important signaling molecules involved in the regulation of the nervous system, immune response, intestinal permeability, and the blood-brain barrier. Various tryptophan derivatives are adsorbed by the intestinal epithelium and enter the bloodstream, influencing different organs, tissues, and physiological functions [13,14,15]. These metabolites, being monocarboxylates, can enter the cytosol by means of a monocarboxylate carrier present in all cells and mitochondria [20]. In contrast, the penetration of itaconic acid, which is a dicarboxylate, into, or its exit out of, cells is questionable. For example, the respiration of intact neurons was not affected by itaconate, but permeabilized cells, as well as isolated brain mitochondria, demonstrated decreased rates of respiration in the presence of itaconate [19]. It can be assumed that acidosis can promote the penetration of itaconic acid via a monocarboxylate transporter as happens with the transport of succinate [21]. In this aspect, the removal of acidosis may be important, among other things, to reduce the permeability of cell membranes.

As shown earlier, itaconate supplementation in permeabilized cells significantly reduced respiration in the presence of succinate and did not affect oxygen consumption in the presence of either pyruvate with malate or glutamate with malate, substrates that drive respiration via complex I [7]. Our results are in complete agreement with these data; however, they also show that the oxidation of glutamate itself is affected, which is both in the decrease in MTT reduction and the activation of NADH oxidation. According to our data, the decrease of the reduced NADH induced by ITA and ICA is associated with a partial suppression of glutamate dehydrogenase activity. The inhibition of glutamate dehydrogenase activity with a concomitant activation of NADH oxidation is a new property of ITA. Thus, among mitochondrial enzymes, the most accessible ITA targets are not only SDH, but also GDH. This result is important in the context of a participation of different metabolic pathways involved in bacterial inflammation and sepsis, in particular, glutaminase pathway with glutamine formation or the reductive and oxidative pathways with accumulation of citrate or succinate, respectively. The involvement of glutamine synthase is supported by data on a more than a threefold increase of the conversion of glutamate to glutamine in sepsis [22]. Also, myocardial levels of glutamate and glutamine were higher in LPS-induced septic shock than in the control [23]. The oxidative pathway is associated with redox-dependent reactions; this is indicated by the fact that NADH oxidation induced by ITA and ICA is eliminated by the lipid radical scavenger BHT. This effect may be due to the oxidative stress associated with the partial inhibition of the complexes of the respiratory chain by itaconate, shown in several studies [19,24].

While BHT and DTT are well known as protectors against oxidative stress, DTT, in addition to this action, is able to influence the interaction of ITA with the thiol groups of enzymes. This interaction underlies the ITA-induced inhibition of several glycolytic enzymes, shown in a number of studies [25,26]. In addition, some effects of ITA decreased in the presence of cysteine [27]. Moreover, ITA could directly modify cysteine sites on functional proteins which related to inflammation [28].

Alkalization has also proven to be effective against many mitochondrial disturbances. As we have shown earlier, an increase in pH, even in a small range, sharply increased the resistance to oxidants [18] and prevented the opening of MPTP [17]. According to our data, alkalization removed almost all the effects of ITA and ICA. The exceptions were the inhibition of DCPIP reduction caused by ITA associated with the direct inhibition of the catalytic subunit of SDH, and a decrease in NAD reduction during oxidative phosphorylation by both. 

It can be assumed that several factors may be involved in the pH-dependent action of inhibitors on the enzyme activity. First, acidification can alter the binding of both the substrate and the inhibitor to the catalytic subunit, probably in favor of the inhibitor. Second, the ability of acceptors to bind electrons can also change depending on pH. Third, the transfer of electrons from the catalytic subunit to the acceptor may undergo inhibition. The latter assumption is supported by the results on strong changes in reactions with the intermediate electron carrier PMS, which are especially pronounced under the influence of alkalization. These aspects require an additional clarification.

In all experiments involving the measurements of MTT reduction, the membrane potential, and NADH oxidation, the effective concentrations of ICA were one order of magnitude lower compared to that of ITA. Another indolic acid, IAA, acted in a similar manner, but weaker than ICA. Their influence increased significantly with medium acidification. As we have shown earlier, phenolic acids, the microbial metabolites of phenylalanine, also disturbed the mitochondrial function. Their effects on the succinate oxidation and nonspecific membrane permeability strongly increased with acidification, substrate deficiency and loading with calcium and iron ions, i.e., conditions associated with bacterial inflammation and sepsis [5,29]. In addition, as lactic acidosis increased, the levels of succinate, 2-oxoglutarate, and fumarate in the blood of patients with sepsis also increased, indicating mitochondrial dysfunction [5]. Unlike these conventional metabolites, ITA was detected in the blood only at an early stage of infection. All of them can affect mitochondrial functions, probably acting at different stages of infection and sepsis, especially in the center of inflammation.

The scheme (Figure 8) summarizes our data on the influence of microbiota-related metabolites, including LPS, on succinate, glutamate and NADH oxidation. LPS provokes the formation of ITA in the tricarboxylic acid cycle of mitochondria of immune cells. ICA is a typical microbial metabolite of tryptophan degradation. Both metabolites inhibit succinate and glutamate oxidation, thus shifting the redox potential of pyridine nucleotides to a more oxidized state. Acidosis enhances these effects, while the thiol antioxidant DTT and the scavenger of lipid radicals BHT partially eliminate them. The alkalization of the medium is the most effective means of protecting mitochondria from these microbiota-related metabolites.

In the clinical aspect, the development of metabolic acidosis is characteristic of the progression of infectious and inflammatory diseases, septic conditions, and multiple organ failure. Until now, lactic acidosis remains one of the main indicators of the severity and prognosis of sepsis [30,31]. It is believed that with the development of acidosis, homeostasis is disturbed, and the effect of therapy is often reduced or even distorted as a result of the disruption of enzymatic processes in organs and cells [32]. 

These events can occur especially intensively in the area of an infectious organ or tissue damage, where metabolites can concentrate due to the accumulation of bacteria and macrophages as their sources. Both ITA and ICA have similar effects on mitochondria, but ICA acts at concentrations an order of magnitude lower than ITA. Besides, ICA and other indolic acids are normally present in the intestine, where the environment is alkaline, which nullifies their effect. As we have shown earlier, ITA is detected in the blood at the early stages of sepsis and disappears at subsequent stages, while the level of microbial metabolites, on the contrary, increases along with the development of acidosis and the severity of pathology [5]. As shown, the inhibition of SDH by itaconate was reversible and occurred within seconds; so, SDH might be an early target of itaconate to modify metabolism [7,19]. It is assumed that the rise of ITA in the period of excessive inflammation and its decline in the immunoparalysis phase can effectively control the progression of sepsis [19,33].

The results of our experiments show that both microbial metabolites ITA and ICA can influence the metabolism of glutamate, suppressing the activity of glutamate dehydrogenase. Since the inhibition of GDH by microbial metabolites should lead to the accumulation of glutamate, a change in the level of glutamate can also be considered as an indicator of inflammation and septic processes. In this regard, recent metabolomic studies aimed at finding new biomarkers of sepsis based on the amino acid profile are of great interest. It was found that the amino acid profile of blood serum in patients with sepsis differs significantly from that in healthy people and patients with inflammation only [22,34,35]. As these studies have shown, the plasma concentration of most of the amino acids was lower in critically ill patients with significant reductions, over 30%, in plasma glutamate and glutamine. Also, a more than threefold increase in the conversion of glutamate to glutamine was found in sepsis [22]. Previously, a decrease in the concentration of glutamate in the blood was found in early phases of septic shock with acute liver dysfunction, which correlated with mortality [35]. Conversely, the myocardial levels of glutamate and glutamine were higher in LPS-induced septic shock than in controls. Besides, in a model of the experimental sepsis, elevated glutamate and glutamine levels in the brain frontal cortex were found [36]. All these data indicate disturbances in glutamate homeostasis both in the tissue and plasma during inflammation and sepsis. Thus, glutamate can serve as an additional marker of these pathologies, and probably as a specific indicator for different stages of their progression.

Alkaline therapy is frequently used in acidosis, which increases with the progression of sepsis and other severe pathologies. As current data have shown, alkaline therapy protects renal function, suppresses inflammation, and improves cellular metabolism in kidney disease [37,38]. It has been suggested that the targets of this therapy are mainly associated with lipid metabolism and oxidoreductase activities, which is in full agreement with our data. It can be noted that the data obtained on the risk of mitochondrial dysfunction progression under the negative influence of microbial metabolites in conditions of acidosis confirm the vital importance of empirically developed tactics of the continuous laboratory monitoring of the acid–base state and its dynamic correction in critically ill patients.

## 4. Materials and Methods

### 4.1. Reagents and Chemicals

All reagents were from the Sigma–Aldrich Corporation (St. Louis, MO, USA).

### 4.2. Preparation of Rat Liver Mitochondria

Mitochondria were isolated from adult Wistar male rats. The study was conducted in accordance with the ethical principles formulated in the Helsinki Declaration on the care and use of laboratory animals. Manipulations were carried out by the certified staff of the Animal Department of the Institute of Theoretical and Experimental Biophysics (Russian Academy of Science) Pushchino, Moscow region, and approved by the Commission on Biomedical Ethics of ITEB RAS (N 2/2022 from (5 March 2022)). During the study, the animals were kept in wire-mesh cages at room temperature with a light/dark cycle of 12 h. Mitochondria from the liver were isolated using the standard method [39,40]. The liver was rapidly removed and homogenized in an ice-cold isolation buffer containing 300 mM sucrose, 1 mM EGTA, and 10 mM HEPES–Tris (pH 7.4). The homogenate was centrifuged at 600× *g* for 7 min at 4 °C and the supernatant fraction was then centrifuged at 9000× *g* for 10 min to obtain mitochondria. Mitochondria were washed twice in the above medium without EGTA. The final mitochondrial pellet was suspended in the washing medium to yield 60 mg protein/mL and kept on ice for analysis.

### 4.3. Determination of Succinate Dehydrogenase Activity from Reduction of Methyl Thiazolyl Tetrazolium (MTT) and Dichlorophenolindophenol (DCPIP)

An incubation medium (2 mL) containing 125 mM KCl, 20 mM HEPES, pH 7.4, 150 µM MTT, and the oxidation substrate were mixed with mitochondria (0.5 mg protein per mL) and incubated for 5 min as described earlier [40]. The examined samples were placed simultaneously in a series of spectrophotometric cuvettes. The reaction of acceptor reduction was initiated by addition of mitochondria. Subsequent to the incubation, mitochondria were lysed by Triton X-100 (10 µL of 10% solution) and optical density was recorded at 580 nm with an USB4000 spectrophotometer (Ocean Optic, Dunedin, FL, USA).

The activity of SDH was also determined by the reduction of the electron acceptor DCPIP [41]. Mitochondria (0.5 mg protein/mL) were incubated in 2 mL of a medium containing 125 mM KCl, 15 mM HEPES, pH 7.4 in the presence of 1 mM cyanide, 10 µL of 10% Triton X-100, 200 µM DCPIP. The DCPIP reduction was induced by the addition of 5 mM succinic acid, and further was activated with 250 µM PMS. The acceptor reduction rate was measured at a wavelength of 600 nm using an Ocean Optics USB4000 spectrophotometer.

### 4.4. Determination of Mitochondrial Membrane Potential

The difference in the electric potential on the inner mitochondrial membrane was measured from the redistribution of lipophilic cation tetraphenylphosphonium (TPP+) between the incubation medium and mitochondria [17,29]. The concentration of TPP+ in the incubation medium was recorded by a TPP+ selective electrode (Nico, Moscow, Russia).

### 4.5. Determination of the Redox State of Pyridine Nucleotides and Oxidative Phosphorylation in Mitochondria

The redox state of pyridine nucleotides and oxidative phosphorylation in the mitochondria in suspension was determined by the fluorescence of pyridine nucleotides (excitation at 340 nm, emission at 460 nm) recorded on a Hitachi-F700 fluorimeter (Japan), as described earlier [39]. Mitochondria (0.6 mg protein/mL) were added to the medium containing 125 mM KCl, 1.5 mM KH2PO4 and 15 mM HEPES–Tris (pH 7.25), as well as glutamate, malate, KGL, pyruvate (2 mM) as the substrates of oxidation. ADP (100 μM) was added to evaluate the oxidative phosphorylation. Complete oxidation of pyridine nucleotides was induced by the addition of an uncoupler FCCP (0.5 μM). The activity of NAD-dependent dehydrogenases was estimated from the reduction of exogenous NAD in permeabilized mitochondria. The experiments were carried out in the in mitochondria permeabilized by a single freezing-thawing. Under these conditions, NAD penetrated across mitochondrial membranes and was reduced by matrix dehydrogenases.

### 4.6. Statistical Analysis

The data given represent the means ± standard error of means (SEM) from five to seven experiments, or are the typical traces of three to five identical experiments with the use of different mitochondrial preparations. Statistical significance was estimated by the Student’s *t*-test with *p* < 0.05 as the criterion of significance.

## Figures and Tables

**Figure 1 ijms-23-09069-f001:**
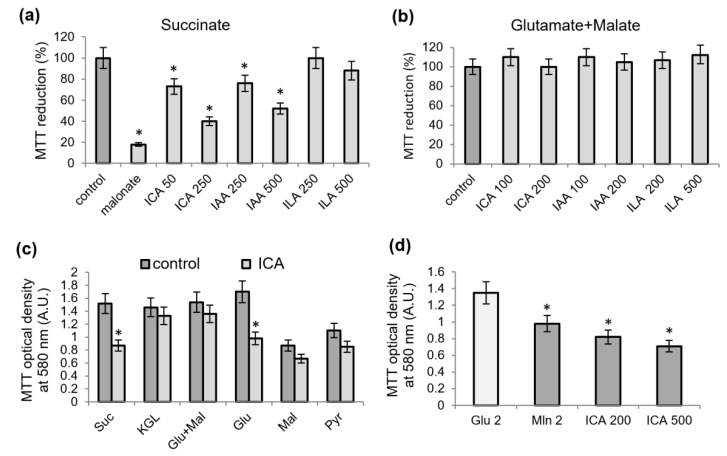
Influence of indolic acids on the oxidation of mitochondrial substrates, as measured by the MTT assay. MTT reduction (%) after a 5-min incubation of mitochondria with indolcarboxylic acid (ICA, 50 and 250 µM), indolacetic acid (IAA, 250 and 500 µM) and indollactic acid (ILA, 250 and 500 µM), and succinate (**a**) or glutamate plus malate (**b**) as substrates of oxidation; influence of ICA (200 µM) on MTT reduction after a 5-min incubation with mitochondria during the oxidation of succinate and NAD-dependent substrates (**c**); effects of malonate (2 mM) and ICA (200 and 500 µM) on MTT reduction activated by glutamate (**d**). Substrates were added at a concentration of 2 mM. Asterisks (*) indicate values that differ significantly from the control values (*p* < 0.05).

**Figure 2 ijms-23-09069-f002:**
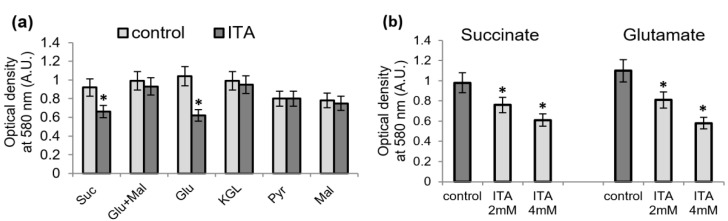
Influence of itaconic acid on the oxidation of mitochondrial substrates, as measured by MTT reduction. Influence of itaconic acid (ITA, 2 mM) on MTT reduction after a 5-min incubation with mitochondria upon the oxidation of succinate and NAD-dependent substrates (**a**); inhibition of MTT reduction by different concentrations of ITA upon the oxidation of succinate (2 mM) or glutamate (2 mM) (**b**). Asterisks (*) indicate values that differ significantly from the control values (*p* < 0.05).

**Figure 3 ijms-23-09069-f003:**
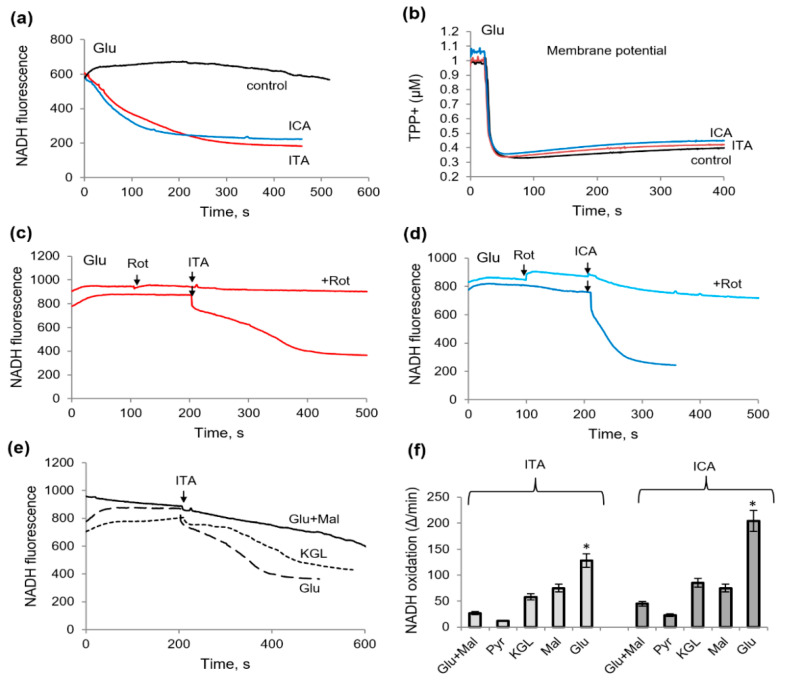
Influence of itaconic and indolcarboxylic acids on the oxidation of glutamate and other NAD-dependent substrates, as measured by NADH fluorescence assay. The activation of NADH oxidation by ICA (250 µM) and ITA (2 mM) in the presence of 2 mM glutamate (**a**); ITA and ICA do not influence the membrane potential supported by glutamate (**b**); the elimination of NADH oxidation induced by ITA (**c**) and ICA (**d**) by rotenone; the influence of ITA on NADH oxidation in the presence of glutamate plus malate, 2-oxoglutarate, and glutamate (**e**); influence of ITA (2 mM) and ICA (250 µM) on NADH oxidation rate in the presence of different NAD-dependent substrates (**f**). Asterisks (*) indicate values that differ significantly from the control values (*p* < 0.05).

**Figure 4 ijms-23-09069-f004:**
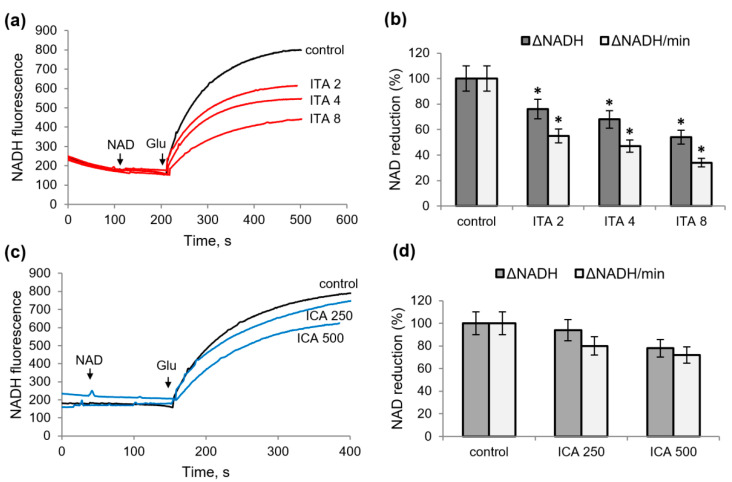
Influence of itaconic and indolcaboxylic acids on the reduction of NAD by glutamate in permeabilized mitochondria. Influence of ITA on the reduction of NAD (0.5 mM) (**a**) and the amplitudes and rates of NAD reduction (**b**) by glutamate; influence of ICA on the reduction of NAD (0.5 mM) (**c**) and the amplitudes and rates of NAD reduction (**d**) by glutamate. Asterisks (*) indicate values that differ significantly from the control values (*p* < 0.05).

**Figure 5 ijms-23-09069-f005:**
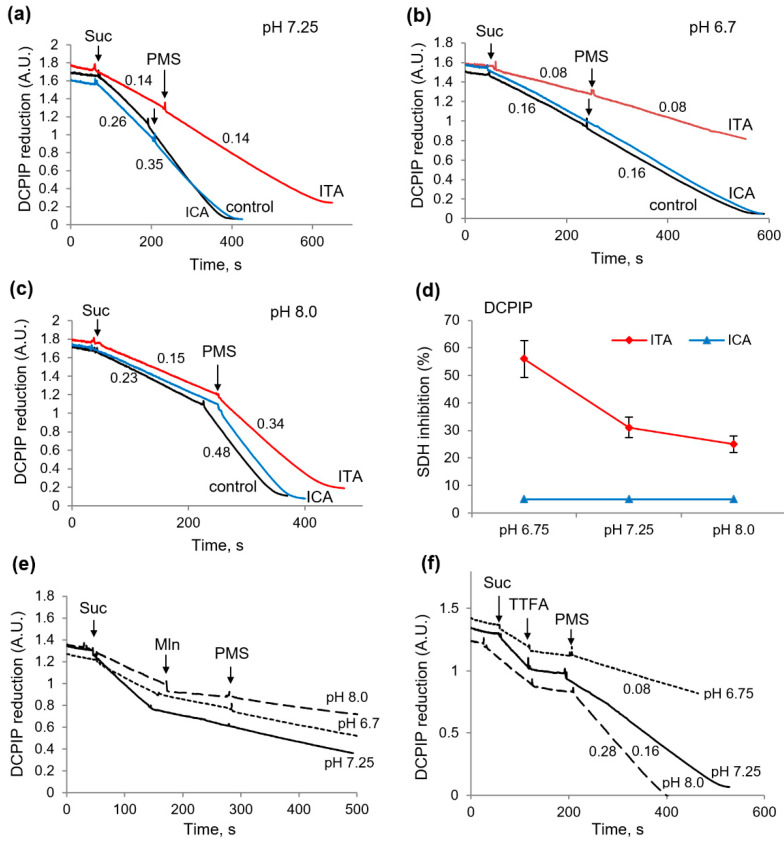
Influence of itaconic and indolcaboxylic acids on the activity of SDH, as measured by DCPIP reduction. DCPIP reduction in the course of succinate (2 mM) oxidation in the presence of ITA (2 mM) and ICA (250 µM) at medium pH equal to 7.25 (**a**), 6.7 (**b**), and 8.0 (**c**), and SDH inhibition (%) at different pH values (**d**). The influence of the inhibitors of SDH malonate (2 mM) (**e**) and TTFA (200 µM) (**f**) on DCPIP reduction and activation by PMS (250 µM) at different pH values of medium; the rates of DCPIP reduction, ΔD/min, are indicated in parentheses.

**Figure 6 ijms-23-09069-f006:**
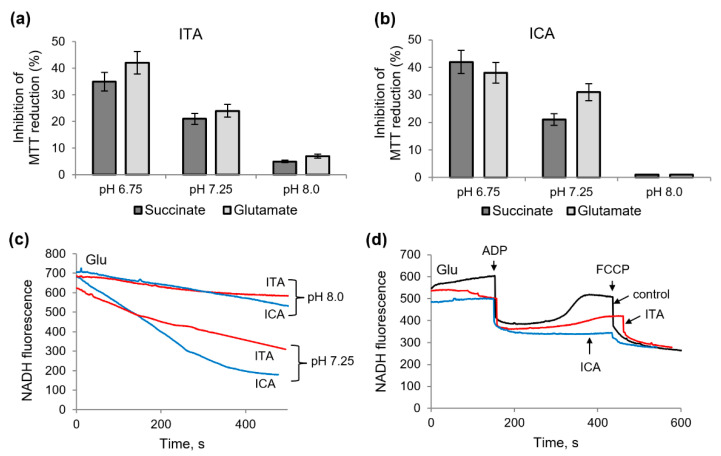
Elimination of the influence of itaconic and indolcaboxylic acids on the MTT reduction and NADH oxidation by the alkalization of medium. The influence of the pH of medium) on MTT reduction during succinate and glutamate oxidation in the presence of 2 mM ITA (**a**) and 250 µM ICA (**b**); the elimination of NADH oxidation, induced by ITA and ICA, by alkalization (**c**) and changes in NAD/NADH redox states during oxidative phosphorylation in the presence of ITA and ICA (**d**) at pH 8.0; additions: 200 µM ADP and 0.1 µM FCCP.

**Figure 7 ijms-23-09069-f007:**
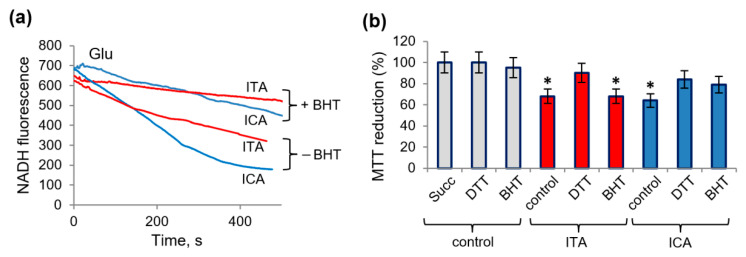
Elimination of the influence of itaconic and indolcaboxylic acids on NADH oxidation and MTT reduction by the antioxidants BHT and DTT. The influence of butyl hydroxytoluene (BHT, 20 µM) on NADH oxidation induced by ITA and ICA in the presence of glutamate as a substrate (**a**); MTT reduction (%) after the incubation of ITA and ICA with mitochondria in the presence of BHT (20 µM) or DTT (2 mM) and succinate as a substrate (**b**). Asterisks (*) indicate values that differ significantly from the control values (*p* < 0.05).

**Figure 8 ijms-23-09069-f008:**
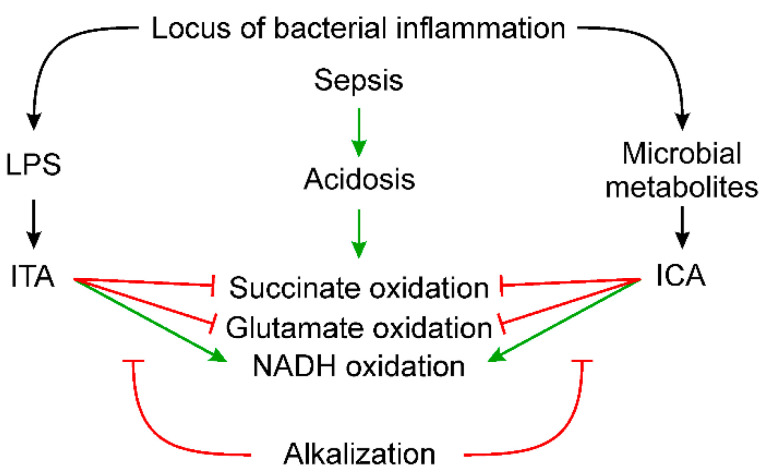
Influence of LPS and microbial metabolites on succinate, glutamate and NADH oxidation

## Data Availability

The datasets generated during and/or analyzed during the current study are available from the corresponding author on reasonable request.

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
