# Peer review of "Influence of Microbial Metabolites and Itaconic Acid Involved in Bacterial Inflammation on the Activity of Mitochondrial Enzymes and the Protective Role of Alkalization"

_ijms, 2022, doi:10.3390/ijms23169069_

Round 1

Reviewer 1 Report

The authors described the effects of indolic acid or ITA, which are microbiota-related metabolites, preciously using purified mitochondria in vitro. These results are interesting enough and keep enough potential in prospects. However, results are only shown by purified mitochondria assay system with no cell-based assays or experimental animal-based assays, indicating that we can’t assess the effects of indolic acid, ITA, or alkalization upon cellular effects or individual effects. Since the cellular or physical homeostasis is adjusted precisely, I can’t believe these results of alkalization in vitro correspond with in vivo, several cell-based assays must be required. If authors describe the effects of these compounds biochemically, more directs in vitro experiments using purified succinate dehydrogenase and glutamate dehydrogenase must be necessary.

Author Response

We agree with the Reviewer that the effects obtained on isolated mitochondria could not always be realized in vivo. Moreover, we ourselves encountered once such a case in our studies. Therefore, we first detect the targets and determine the conditions in which the action of microbial metabolites is most pronounced. As we have shown earlier, these conditions include substrate deficiency and acidosis, which accompany inflammation. This is an important aspect because mitochondrial dysfunction is now considered to be one of the main reasons for multiple organ failure in bacterial inflammation and sepsis.

Our data show that indolic acids inhibit the oxidation of succinate and glutamate in intact mitochondria. They do not directly affect the activity of SDH and GDH in the standard assays of enzyme activity in lysed mitochondria.  This experimental finding distinguishes their action from the influence of itaconic acid, which inhibits these enzymes in the same tests. Obviously, these differences can be revealed only when using isolated mitochondria as an object for study, but not cells or purified enzymes.

Besides, the study of microbial metabolites in vivo would be associated with a number of fundamental limitations. First of all, this refers to the permeability of cell membranes to these metabolites. Indolic acids, being monocarboxylates, can enter the cell by means of a monocarboxylate carrier present in all cells. In contrast, the penetration of itaconic acid, which is a dicarboxylate, into cells is questionable. These circumstances do not limit the study of microbial metabolites on mitochondria since mitochondria contain both monocarboxylate and dicarboxylate transporters.

For the above reasons, to date we can assume that the detected effects of microbial metabolites and itaconate are realized mainly in the foci of inflammation, where bacteria, macrophages and their metabolites are concentrated. If we would manage to identify a microbial metabolite that can simultaneously serve as a biomarker for inflammation and sepsis, then it will be appropriate to conduct in vivo studies.

Reviewer 2 Report

The manuscript presented by Fedotcheva and Beloborodova describes the study on the effects of itaconate acid and indole acids on mitochondrial enzymes. In particular, the authors investigate the inhibition by itaconate and indolcarboxylic acid (ICA) of succinate dehydrogenase, glutamate dehydrogenase and mitochondrial oxidation of NADH.

The manuscript reports data about well-known effects of itaconic acid on succinate dehydrogenase. Instead, new data concern the effect of indolcarboxylic acid on mitochondrial enzymes. The manuscript needs revision to facilitated reading. The title of manuscript refers to generic “Microbiota-Related Metabolites”, but the authors used itaconate, produced by human immune blood cells, and indolcarboxylic acid, derived from the microbial pathway of tryptophan degradation, in their experiments. Furthermore, no direct data on bacterial inflammation are reported. Based on these observations I suggest modifying the title referring only to the effect of ICA on mitochondrial enzymes, without indicating the effect on enzymes in bacterial inflammation.

Maior revisions

-    Have the authors tested ITA inhibition at 200-500uM? Indicate in text if the authors did it.

-     Figure 4b and 4D, lane 158-161:   The authors report the decrease in the reduction of NAD, but in the figure 4 statistical analysis is missing. Were the differences statistically significant?

-    Lane 176-179: Acidification reduce also the SDH activity in control samples. The reduction of SDH activity by ITA at pH 6.7 is more evident than at pH 7.25. This point should be discussed.

-     Figure 7B: no statistical analysis is reported

 Minor revisions

The list of abbreviations should be added.

-          In the abstract acronyms and abbreviations should be defined (eg. MTT, DCPIP, BHT, etc.)

-     Figure 2: The * of statistical significance should be put on gray bar (Itaconic inhibition).

Author Response

Replays:

Although the inhibition of SDH by itaconic acid has been studied, we expand the study and show that the inhibition of SDH by itaconic acid is enhanced with acidosis. In addition, we show for the first time that itaconic acid inhibits GDH. Also, the protection by alkalization also relates to the infuence of itaconic acid.

Really, we have used the term Microbiota-Related Metabolites to combine microbial metabolites and itaconic acid in the title of the manuscript. We assume that this combination is consistent with the fact that itaconic acid is formed only in response to the action of bacterial lipopolysaccharides during bacterial inflammation. Nevertheless, we changed the title to a more specific:

It was:

Influence of Microbiota-Related Metabolites on the Activity of Mitochondrial Enzymes in Bacterial Inflammation and the Protective Role of Alkalization

Now it reads as follows:

Influence of Microbial Metabolites and Itaconic Acid Involved in Bacterial Inflammation on the Activity of Mitochondrial Enzymes and the Protective Role of Alkalization

Maior revisions

-    Have the authors tested ITA inhibition at 200-500uM? Indicate in text if the authors did it.

Since ITA is a competitive inhibitor of SDH, we tested ITA in the concentration range of 2-8 mM and concentration of succinate of 1-2 mM.  To test low concentrations of ITA, it would be necessary to lower the concentration of succinate, which greatly decreases the control values of the optical density of the reduced acceptors, MTT and DCPIP, and lowers the sensitivity of the assay.

-     Figure 4b and 4D, lane 158-161:   The authors report the decrease in the reduction of NAD, but in the figure 4 statistical analysis is missing. Were the differences statistically significant?

We have added asterisks to the figure.

-    Lane 176-179: Acidification reduce also the SDH activity in control samples. The reduction of SDH activity by ITA at pH 6.7 is more evident than at pH 7.25. This point should be discussed.

At present, we think that several factors may be involved in these pH-dependent changes in enzyme activity. First, acidification can alter the binding of both the substrate and inhibitor to the catalytic subunit, probably in favor of the inhibitor. Second, the ability acceptors to accept electrons can also change depending on pH. Third, the transfer of electrons from the catalytic subunit to the acceptor may be subject to inhibition. The latter factor is supported by results on strong changes in reactions with the intermediate electron carrier PMS. This aspect requires a large number of additional experiments with different specific inhibitors for clarification.

We have added the following paragraph to the Discussion section:

It can be assumed that several factors may be involved in the pH-dependent action of inhibitors on the enzyme activity. First, acidification can alter the binding of both the substrate and the inhibitor to the catalytic subunit, probably in favor of the inhibitor. Second, the ability of acceptors to bind electrons can also change depending on pH. Third, the transfer of electrons from the catalytic subunit to the acceptor may undergo inhibition. The latter assumption is supported by the results on strong changes in reactions with the intermediate electron carrier PMS, which are especially pronounced under the influence of alkalization. These aspects require an additional clarification.

-     Figure 7B: no statistical analysis is reported

We have added asterisks to the figure.

 Minor revisions

The list of abbreviations should be added.

-          In the abstract acronyms and abbreviations should be defined (eg. MTT, DCPIP, BHT, etc.)

We have added a definition of abbreviations to the abstract

-     Figure 2: The * of statistical significance should be put on gray bar (Itaconic inhibition).

We moved asterisks to the required columns

Reviewer 3 Report

I know the authors from the "Tetrazolium dye"  paper (2017). Good ideas,  because Microbiota-related metabolites have a direct relation to clinical research /translational research. I would appreciate if the group could present oxymetry data from mitochondria. But this means a suggesion, only. It does not lessen the value of the paper. Congratulations!

Author Response

Thanks for supporting our research. As for a mitochondrial respiration, we plan to measure this integral function in further experiments, when we identify specific mitochondrial targets of microbial metabolites.
